# MEASURING AND CONTROLLING SOLUTION DEGENERACY ACROSS TASK-TRAINED RECURRENT NEURAL NETWORKS

## ABSTRACT

Task-trained recurrent neural networks (RNNs) are versatile models of dynamical processes widely used in machine learning and neuroscience. While RNNs are easily trained to perform a wide range of tasks, the nature and extent of the degeneracy in the resultant solutions (i.e., the variability across trained RNNs) remain poorly understood. Here, we provide a unified framework for analyzing degeneracy across three levels: behavior, neural dynamics, and weight space. We analyzed RNNs trained on diverse tasks across machine learning and neuroscience domains, including N-bit flip-flop, sine wave generation, delayed discrimination, and path integration. Our key finding is that the variability across RNN solutions, quantified on the basis of neural dynamics and trained weights, depends primarily on network capacity and task characteristics such as complexity. We introduce information-theoretic measures to quantify task complexity and demonstrate that increasing task complexity consistently reduces degeneracy in neural dynamics and generalization behavior while increasing degeneracy in weight space. These relationships hold across diverse tasks and can be used to control the degeneracy of the solution space of task-trained RNNs. Furthermore, we provide several strategies to control solution degeneracy, enabling task-trained RNNs to learn more consistent or diverse solutions as needed. We envision that these insights will lead to more reliable machine learning models and could inspire strategies to better understand and control degeneracy observed in neuroscience experiments.

## 1 INTRODUCTION

Recurrent neural networks (RNNs) are widely used across machine learning and computational neuroscience for modeling dynamic processes. They can be efficiently trained using standard nonconvex optimization techniques and have proven useful for understanding neural dynamics during task performance (Sussillo, 2014; Rajan et al., 2016; Barak; Mastrogiuseppe & Ostojic, 2018; Vyas et al., 2020; Driscoll et al., 2024). While it is well-known that in feedforward networks, differences in weight initialization and randomness during training (stochastic gradients, mini-batch sample variability, etc.) can cause different final solutions, less is known about how such differences manifest when training RNNs (Das & Fiete, 2020; Turner & Barak, 2024; Collins et al., 2022; Glorot & Bengio, 2010; Fort et al., 2019; Goodfellow et al., 2015; Li et al., 2018; Jastrzebski et al., 2018; Chaudhari et al., 2017; Frankle & Carbin, 2019; Kornblith et al., 2019).

Traditionally, the study of task-trained RNNs has often focused on analyzing models trained using a single approach, implicitly assuming that multiple RNNs trained on the same task would converge to similar solutions, even when trained differently or when starting from different initial conditions. However, recent work has shown that this assumption may not hold universally. For instance, Maheswaranathan et al. (2019) found that while trained RNNs may share certain topological features, their internal representation geometry can vary widely. Similarly, Turner et al. (2021) discovered that task-trained networks can develop qualitatively distinct dynamics.

These contrasting findings raise a fundamental question about the degeneracy of task-trained RNN solutions: when do networks trained on the same task converge to similar dynamics and representations, and when do they exhibit substantial degeneracy? Previous studies offer conflicting per-

spectives, with some suggesting that task-trained RNNs exhibit dynamics that are universal across instances (Maheswaranathan et al., 2019), while others emphasize the degeneracy across individual solutions (Turner et al., 2021; Galgali et al., 2023; Gholamrezaei & Whishaw; Gao et al.; Mehrer et al.).

In this paper, we reconcile these differing views by providing a unified framework for analyzing degeneracy at three levels: behavior, neural dynamics, and weight space. We hypothesize that the variation in solutions of trained RNNs is influenced by *task under-specification*. In other words, when constraints do not uniquely or adequately determine the network's solution to a given task, we observe greater variability across trained networks (D'Amour et al., 2020). To test this hypothesis, we investigate how different task characteristics, particularly at the level of complexity of the inputs and outputs, affect degeneracy of the solutions found by task-trained RNNs at three levels (across behavioral, neural, and weight space). We simulated four tasks – N-Bits Flip Flip, Delayed Discrimination, Sine Wave Generation, and Path Integration 1. Our key finding is that as task complexity increases, the solution space becomes more constrained, reducing degeneracy in both behavior and neural dynamics but increasing degeneracy in the underlying network weights.

Our simulations confirm the above hypothesis at the behavioral and neural-dynamical levels: with increasing task complexity, the degeneracy of solutions – measured at the behavioral level by the coefficient of variation of the out-of-distribution performance and at the dynamical level by the Dynamical Similarity Analysis distance – decreases consistently across all four tasks we tested 3 5 (Ostrow et al., 2023). Interestingly, at the weight level, we observe the opposite trend: the degeneracy of solutions increases with task complexity.

After quantifying degeneracy at behavioral, neural, and weight-levels, next we propose practical strategies for controlling the level of degeneracy in task-trained RNNs, including altering task complexity, incorporating auxiliary loss functions, and applying structural constraints during training. These methods provide flexibility for researchers aiming to tailor the training process to their specific needs, whether they are seeking more consistent (Kepple et al., 2022) or more diverse RNN solutions (Liebana Garcia et al., 2023; Fascianelli et al., 2024; Pan-Vazquez et al., 2024; Kepple et al., 2022).

## 1.1 CONTRIBUTIONS

Our contributions with this paper are as follows:

- **Information-theoretic quantification of task complexity**: We introduce a new framework for assessing task complexity by quantifying the information content in the input and target output time series of a task. This measure effectively captures the extent to which input-output signals constrain neural dynamics. Furthermore, this measure correlates well with the neural-dynamical degeneracy we observed both across different complexity levels within the same task and across different tasks.

- **Quantifying the degeneracy of solution spaces**: We measure degeneracy at the behavioral, dynamical, and weight levels in populations of trained networks across four different tasks of wide applicability to both ML and neuroscience. Our findings reveal that, across all tasks, increased task complexity leads to lower degeneracy at the behavioral and neural dynamical levels, but concomitantly higher degeneracy at the weight level.

- **Methods for controlling degeneracy**: We propose five techniques for controlling degeneracy in task-trained RNNs, including manipulating task complexity, applying specific regularization during training through auxiliary losses, and structural constraints during training. Notably, we demonstrate that both neural dynamical degeneracy and weight degeneracy can be manipulated to change either in the same ("covariant") or opposite directions ("contravariance"), providing flexibility for researchers to tailor network solutions to their specific needs (Cao & Yamins, 2024; Kepple et al., 2022; Fascianelli et al., 2024; Maheswaranathan et al., 2019; Durstewitz et al., 2023; Gilpin, 2024) .

## 2 METHODS

### 2.1 MODEL ARCHITECTURE

We used discrete-time nonlinear *vanilla* recurrent neural networks (RNNs), where the update rule is defined as: $\mathbf{h}_t = F(\mathbf{h}_{t-1}, \mathbf{x}_t) = \tanh(\mathbf{W}_h \mathbf{h}_{t-1} + \mathbf{W}_x \mathbf{x}_t + \mathbf{b})$, where $\mathbf{h}_t \in \mathbb{R}^n$ is the hidden state at time $t$, $\mathbf{x}_t \in \mathbb{R}^m$ is the input at time $t$, $\mathbf{W}_h \in \mathbb{R}^{n \times n}$ is the recurrent weight matrix, $\mathbf{W}_x \in \mathbb{R}^{n \times m}$ is the input weight matrix, and $\mathbf{b} \in \mathbb{R}^n$ is the bias vector.

A linear readout layer is applied on top of the RNN hidden state to produce the model's prediction at each time step. The RNNs use Backpropagation Through Time (BPTT) as the learning rule where the RNN is unrolled over time, allowing gradients to be computed for each time step in the sequence (Werbos, 1990). All networks are trained using supervised learning via the Adam optimizer, with a learning rate specific to each task determined via hyperparameter tuning (a list of all training-related hyperparameters can be found in Appendix A). For each task, we train 50 networks using different initializations, until they achieve near-asymptotic loss on the test set. For the N-Bit Flip-Flop and Path Integration tasks 1, we use RNNs with 64 hidden units and input/output dimensions appropriate to each task's specific requirements. For the Delayed Discrimination and Sine Wave Generation tasks 1, we use RNNs with 128 hidden units.

### 2.2 TASKS

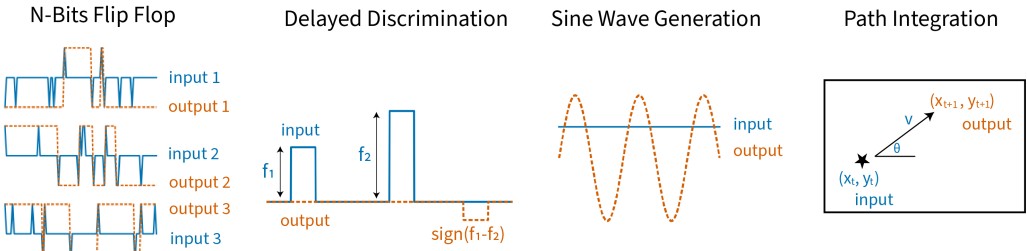

Figure 1: **Task suite:** We train RNNs on a diverse set of four tasks: **N-Bit flip-flop**: Networks are trained to remember the last non-zero input for each of the N channels. **Delayed Discrimination**: Networks compare the magnitude of two temporally separated pulses across N channels. **Sine Wave Generation**: Networks receive static input indicating frequency and produce sine waves of the specified frequency across each of the N channels. **Path Integration**: Networks integrate velocity inputs to track position in a bounded arena in 2D or 3D space. Schematic only shows 2D environment.

**N-Bit Flip-Flop Task** In this task, RNNs are provided with $N$ independent input channels, each taking discrete values from $\{-1, 0, +1\}$, with a fixed probability of switching $p_{switch}$. The network has $N$ output channels, each of which is required to remember the value of the last nonzero input received on its respective input channel. We vary the complexity of the task by changing the number of input and/or output channels $N$.

**Delayed Discrimination Task** The network is presented with two pulses of varying amplitudes $f_1, f_2 \in [2, 10]$, separated in time by a variable delay $t \in [5, 20]$ time steps. The network is required to output the sign of the difference between the two amplitudes, i.e., $\text{sign}(f_2 - f_1)$. This task can be extended to have $N$ independent input and output channels, where the network must compare $f_1$ and $f_2$ within each channel separately. We vary the complexity of the task by varying the number of independent input and/or output channels $N$.

**Sine Wave Generation** RNNs receive a static input indicating a target frequency $f \in [1, 30]$ and are required to generate a sine wave of that frequency over time, effectively converting the input $f$ into the output $\sin(2\pi f t)$. We define $N_{\text{freq}}$ as the number of possible target frequencies presented in the training set, equally spaced within the range $[1, 30]$. This task can involve $N$ independent input and output channels, where each channel outputs a sine wave with the frequency specified by its respective input channel. We vary the complexity of the task by changing $N_{\text{freq}}$ and $N$, the number of independent input and/or output channels.

**Path Integration Task** Each network is initialized at a random initial location within a bounded 2D environment. At each time step, the network receives inputs representing the angular direction $\theta$ and speed $v$, and must integrate this information over time to output the updated $(x, y)$ location. To vary the complexity of the task, we introduce a 3D version of the task, where the network receives inputs $\theta$ (azimuth angle), $\phi$ (elevation angle), and $v$ (speed), and must output the updated $(x, y, z)$ location. The network effectively performs path integration by accumulating movement vectors based on the input directions and speeds.

## 3 RESULTS

### 3.1 CHARACTERIZING TASKS COMPLEXITY AND NEURAL DYNAMICS (RNN HIDDEN STATES)

We quantified the task complexity by calculating the Shannon entropy of its input and target (output) time-series (Shannon, 1948; Bialek et al., 2001; Crutchfield & Young, 1989). We chose this measure because it directly reflects the amount of information present in the task's inputs and outputs, which the network must process and represent through its hidden state (Tishby et al., 2000). By quantifying the statistical complexity of the input and output time series, we capture the information-processing demand imposed on the network.

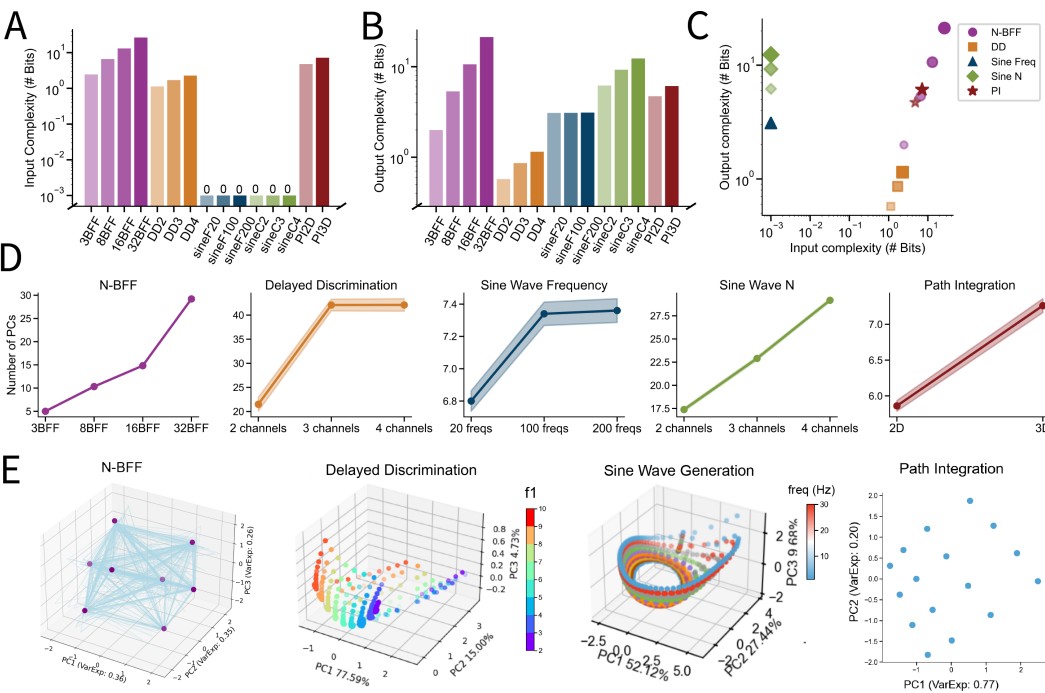

Figure 2: **Characterizing tasks and RNN hidden states (neural dynamics):** We analyze task complexity and its effects on RNN hidden states using multiple measures: (A) Input complexity, measured as the entropy of the time-averaged input signal. We use the integer in the task name to indicate the number of input-output channels for each task except in "sineF", where "F" stands for frequency, and the integer indicates the number of frequencies the network learns to produce. (B) Output complexity, measured as the entropy of the time-averaged output signal (C) Our tasks span three quadrants of the Input and Output Complexity space (excluding low input and low output complexity). (D) The number of leading Principal Components (PCs) explaining 95% of the variance, which we use as a measure of the dimensionality of the task, increases consistently across tasks as difficulty increases. (E) RNN hidden state (neural dynamics) trajectories projected onto the leading two PCs for Path Integration and three PCs for the other tasks.

Without loss of generality, given a one-dimensional output time series $(y_1, y_2, \ldots, y_T)$, the entropy of this time series is given by: $H(Y) = -\sum_{k=1}^{n} p(y_k) \log_2 p(y_k)$, where $n$ is the number of unique values in the time series, and $p(y_k)$ is the probability of observing the value $y_k$ in the time series.

For Sine Wave Generation and Path Integration tasks, the outputs are continuous values rather than discrete ones. Therefore, we binned the output values into discrete intervals before calculating the entropy. For task variants with multiple independent channels, we multiplied the single-channel entropy by the number of channels. Since each channel operates independently and has identical statistical properties, multiplying the single-channel entropy by $N$ provides an accurate measure of the total entropy across all channels. For a training set with $M$ trials, we averaged the entropy across all trials presented in the training set to obtain a representative measure of task complexity:

$$H_{\text{task}} = N \times \left( \frac{1}{M} \sum_{i=1}^{M} H(Y^{(i)}) \right).$$

Different tasks are characterized by their specific input and output complexity profiles (Figure 2A and 2B). Within each task, increasing the number of input and output channels consistently increases both the input and output complexities. We selected our tasks to span three quadrants of the input-output complexity space (Figure 2C):

- High-input, high-output complexity: N-Bit Flip-Flop and Path Integration tasks
- High-input, low-output complexity: Delayed Discrimination task
- Low-input, high-output complexity: Sine Wave Generation task

Among them, the N-Bit Flip-Flop task exhibits the highest input and output complexities among all tasks, suggesting that it imposes the greatest information-processing demands on the networks.

We then provide a link between the task complexity and the dimensionality of the representation demanded by the tasks. We found that the number of leading Principal Components (PCs) explaining 95% of the variance increases consistently across tasks as difficulty increases, indicating that harder tasks demand higher-dimensional representations and fuller utilization of the networks' representational capacity (Figure 2D) (Gao et al.).

Typical solutions found by converged networks across the different tasks are as follows (Figure 2E): In **N-Bit flip-flop** tasks, networks learn two fixed points corresponding to the output value of $\{-1, +1\}$ for each channel. Networks further factorize the fixed points corresponding to different output channels along different orthogonal dimensions. Specifically , when $N = 3$, networks learn $2^3 = 8$ fixed points on the vertices of a 3D cube. In **Delayed Discrimination** tasks, networks memorize the first input value by storing each input value as a separate fixed point during the delay period, when the inputs to the network are no longer on and working memory is required (Hopfield, 1982; Sussillo & Barak, 2013; Driscoll et al., 2024). In **Sine Wave Generation** tasks, networks develop limit cycles corresponding to different target frequencies indicated by inputs, and traverse different limit cycles to produce different sine wave outputs at the appropriate frequencies. In **2D Path Integration** tasks, without any external input, networks develop 2D maps of their environments using a 2D plane of attractors. Without any inputs, a network's hidden state (neural dynamics) stabilizes on fixed points corresponding to their current location.

### 3.2 Characterizing the degeneracy of task-trained RNNs

We now quantify the degeneracy of task-trained RNNs at three levels of analysis: (out of distribution) behavior, neural dynamics, and weight space. We also relate the measured degeneracy back to the task complexity measure introduced in the last section.

#### 3.2.1 Degeneracy in dynamics decreases with task complexity

We use a recently published measure for comparing dynamical systems, called Dynamical Similarity Analysis (DSA) (Ostrow et al., 2023), to perform pairwise comparisons of the neural dynamics in our task-trained networks. DSA measures the temporal and topological structure of the system's

dynamics and ignores geometric configurations that alter the spatial representation of system trajectories without changing the underlying dynamics. It provides a quantitative framework to assess how different networks may arrive at similar or distinct dynamical implementations of a task (See Appendix C for technical details).

To quantify the relationship between task complexity and dynamical degeneracy, we increased the complexity of the tasks by increasing the number of independent input and output channels. This choice is grounded in information theory, as an entropy measure of task complexity is additive for statistically independent variables. We found that within each task, increasing the number of input and output channels consistently reduced the dynamical degeneracy across the converged networks (Figure 3A). Strikingly, this relationship ("covariant" degeneracy with task complexity) holds across all the tasks we consider (Figure 3C).

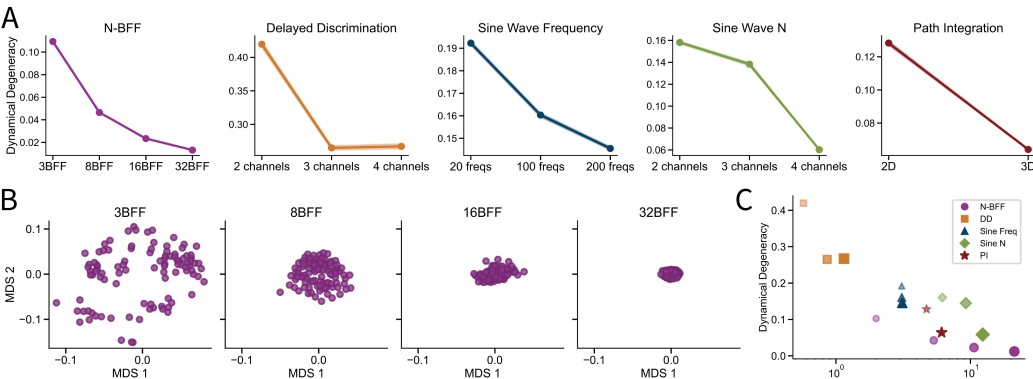

Figure 3: **Dynamical degeneracy decreases with increased task complexity: (A)** Pairwise Dynamical Similarity Analysis (DSA) scores decrease consistently across all tasks as we increase task complexity, suggesting more similar (less degenerate) dynamical solutions for harder tasks. In Sine Wave Frequency, we change the frequency content of the task and in Sine Wave N, we change the number of input-output channels (N) of the task. **(B)** 2D embedding of pairwise DSA distances using Multi-Dimensional Scaling (MDS) on the N-bit flip-flop task shows how the spread of solutions decreases as task complexity is increased. Each dot represents a network with a different initialization. **(C)** Output complexity versus the average dynamical degeneracy on a task shows how higher output complexity correlates with lower dynamical degeneracy both within and across tasks. Within each task, larger marker size and less opacity indicate task variant with higher complexity. Smaller DSA score implies dynamics are more similar.

### 3.2.2 DEGENERACY IN WEIGHTS INCREASES WITH TASK COMPLEXITY

We quantify degeneracy across task-trained RNNs at the level of post-training weights by using a permutation invariant modification of the Frobenius Norm. For a pair of RNNs with recurrent weight matrices given by $\mathbf{W}_1$ and $\mathbf{W}_2$, we define: $d_{\text{PIF}}(\mathbf{W}_1, \mathbf{W}_2) = \min_{\mathbf{P}_1, \mathbf{P}_2 \in \mathcal{P}(n)} \|\mathbf{W}_1 - \mathbf{P}_1 \mathbf{W}_2 \mathbf{P}_2\|_F$, where $\mathcal{P}(n)$ is the set of permutation matrices of size $n \times n$, and $\|\cdot\|_F$ denotes the Frobenius norm. (See Appendix D for additional details) For comparing networks of different sizes, we normalize the above norm by the number of parameters in the weight matrix. We found that pairwise distances between weight matrices from converged RNNs increases consistently as task complexity increases across all tasks (Figure 4A).

Inspired by research linking statistical mechanics of random Gaussian landscapes to deep learning theory (Bahri et al., 2020; Fyodorov & Williams, 2007; Bray & Dean, 2007), we hypothesized that increasing task complexity while holding the network size constant could make possible solutions 'rarer' at the level of recurrent weights. This implies that the optimization process would have to, on average, search for solutions further away from the initial recurrent weights to converge. We found that the distance between the converged weight matrices and the initialized weight matrices indeed increases consistently with task complexity, and does so across all the tasks (Figure 4B and Table 1). Formally, distance from initial weights is $\|\mathbf{W}_T - \mathbf{W}_0\|_F$, where $\mathbf{W}_T$ is the weight matrix

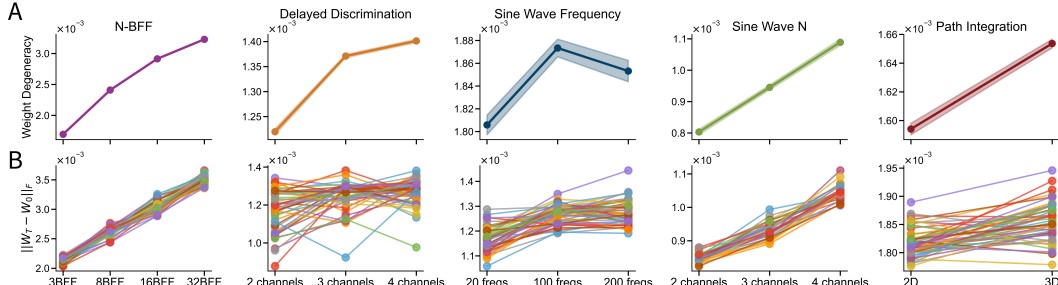

Figure 4: **Degeneracy in weight space (measured using permutation-invariant metrics) increases with increased task complexity**: **(A)** Pairwise distances between weights, measured by the normalized Frobenius norm of the recurrent connectivity matrix , increase consistently with task complexity, across all tasks. **(B)** Distances from initialized weights, quantified by the Frobenius Norm, also increases with task complexity, indicating greater divergence in weight space for harder tasks. In other words, we observe greater variation/variability across trained individuals (individual differences) at the connectivity level, even with the opposite trend observed at the dynamical and behavioral levels. Each line represents the distance from the initialized weights for a given network.

after training and $\mathbf{W}_0$ is the initial weight matrix. We normalize this measure by the number of parameters in the weight matrix across tasks.

### 3.2.3 DEGENERACY IN OUT-OF-DISTRIBUTION GENERALIZATION BEHAVIORS DECREASES WITH TASK COMPLEXITY

Does degeneracy in Out-of-Distribution generalization behaviors change concomitantly with degeneracy in neural dynamics (hidden state) with task complexity? We hypothesize that lower dynamical degeneracy implies that networks respond to input stimuli with more similar dynamics. Therefore, when these networks are probed with out-of-distribution (OOD) inputs, they should produce more similar outputs, leading to lower dispersion in their OOD performance across networks (i.e., lower degeneracy). In other words, as task difficulty increases and dynamical degeneracy decreases, the degeneracy in OOD performance should also decrease.

To test this hypothesis, we measured the OOD performance (mean squared error) of converged networks that all achieved near-asymptotic training loss under the following *length generalization* conditions. For Delayed Discrimination tasks, we doubled the length of the delay period; for all other tasks, we doubled the length of the entire trial. We then calculated the coefficient of variation (CV) of the OOD performance across networks, defined as $CV = \sigma/\mu$, where $\sigma$ is the standard deviation of the OOD performance across networks, and $\mu$ is the mean OOD performance.

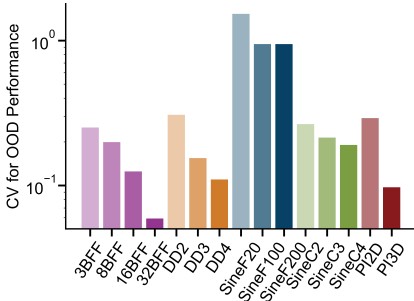

Figure 5: **Behavioral degeneracy, in terms of variability of the OOD generalization performance, decreases with increased task complexity:** Degeneracy of the out of distribution (OOD) generalization behavior, as measured by coefficient of variation of the OOD performance (mean squared error), decreases consistently with increasing task complexity across all tasks.

Our results show that across all tasks, networks with lower dynamical degeneracy indeed exhibited lower CV in their OOD performance 5. This finding supports our hypothesis that increased task complexity, which reduces dynamical degeneracy, also leads to more consistent – less degenerate – OOD generalization behaviors across networks. The prediction this result makes is that when trained on harder tasks, "expert" animals in neuroscience labs or task-trained networks/agents in ML should show less individual variation in generalization performance when tested.

## 3.3 CONTROLLING DEGENERACY OF TASK-TRAINED RNNS

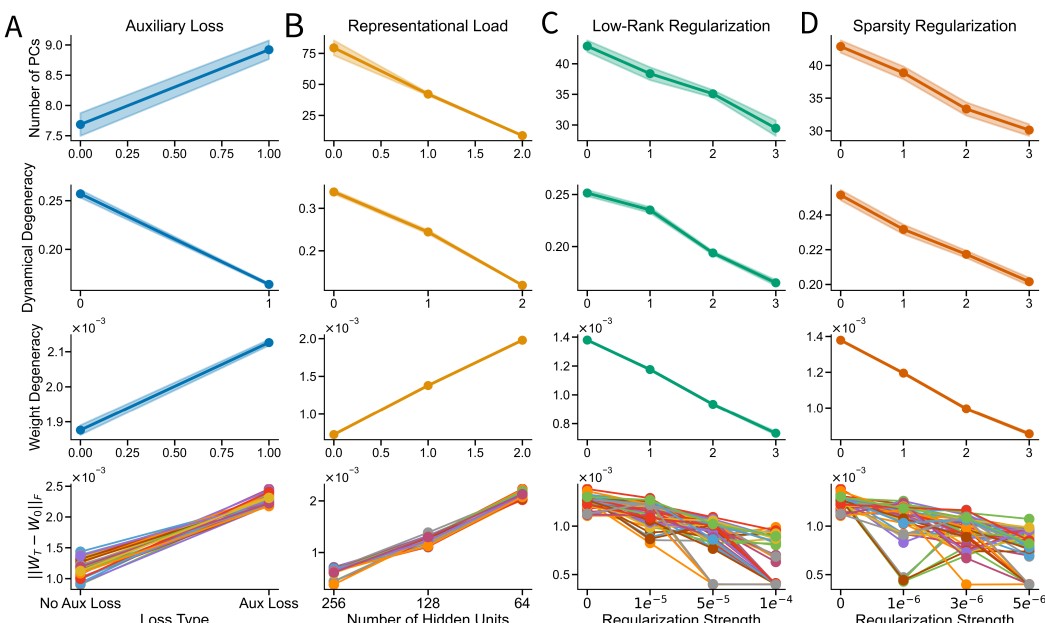

Figure 6: **Differential *control* of weight and dynamics degeneracy:** Using Delayed Discrimination (DD) tasks as an example, we demonstrate various methods to control the degeneracy of solution spaces found by task-trained RNNs: **(A) Adding an auxiliary loss**. Adding an additional loss term to calculate the signed difference (not just direction) between the two consecutive inputs adds additional structure to the learned dynamics, resulting in reduced dynamical degeneracy and increased weight degeneracy. **(B) Changing network size**: By increasing the networks' size, we decrease the load on their representational capacity, which in turn, results in more degeneracy in their neural dynamics or hidden states. Simultaneously, we see greater degeneracy in their weight space (i.e., the trend in degeneracy is in the same direction for both dynamics and weights). **(C) Regularizing weights to be low rank**: We see a decrease in *both* dynamical and weight-space degeneracy with increasing low-rank regularization on the recurrent weights and reduced distance from initialized weights, likely due to smaller-magnitude weight updates. **(D) Regularizing weights to be sparse**: Similar to above, we see a decrease in both dynamical and weight degeneracy with increasing sparsity of the recurrent weight matrix.

Table 1: Impact of Various Factors on Degeneracy

|  | Dynamical Degeneracy | Weight Degeneracy |
|---|:---:|:---:|
| Increase Task Complexity | ↓ | ↑ |
| Auxiliary Loss | ↓ | ↑ |
| Representational Load | ↓ | ↑ |
| Low-Rank Regularization | ↓ | ↓ |
| Sparsity Regularization | ↓ | ↓ |

Based on our characterization of task complexity and quantification of degeneracy at multiple levels of analysis, we propose several methods to control degeneracy across task-trained RNNs, making predictions for ML and neuroscience studies:

**Increase task complexity:** As discussed in the previous section, increasing task complexity reduces dynamical degeneracy and increases weight degeneracy in a consistent and measurable way in a suite of tasks of broad interest to ML and neuroscience.

**Adding auxiliary loss:** We can directly change the degeneracy of the solution spaces found by task-trained RNNs by adding additional terms to the objective/loss function that are consistent with but distinct from the primary task objective. This introduces additional structure to the learned dynamics, which in turn, decreases dynamical degeneracy and increases weight degeneracy, as networks must satisfy multiple constraints simultaneously. Figure 6A shows an example implementation of this idea on a Delayed Discrimination task. We demand that the network output $f_1 - f_2$, instead of only $sign(f_1 - f_2)$. By analyzing the RNN's neural dynamics or hidden state after both input stimuli have been presented, , we find that the additional objective forces the network to learn "line attractors" (Strogatz, 2000) with the same sign as network without the additional loss, but with variable magnitudes reflecting different values of $f_1 - f_2$ (Figure 7).

**Modifying representational load (or network capacity):** Changing the network size (or the number of units in the RNN) affects its representational capacity. Increasing network size typically leads to increased dynamical degeneracy due to more degrees of freedom. Simultaneously degeneracy increases across weight space because of more parameters. (Figure 6 B).

**Imposing structural constraints on weight spaces(or regularization of recurrent weight matrices):** Applying regularization techniques such as low-rank (nuclear norm) (Figure 6 C) or sparsity (L1) (Figure 6 D) regularization to the recurrent weight matrix generally decreases the degeneracy across neural dynamical and weight spaces simultaneously (Mastrogiuseppe & Ostojic, 2018; Narang et al., 2017; Herbert & Ostojic, 2022).

These methods provide a toolkit for controlling degeneracy that naturally follows from our earlier analyses. Importantly, we report that some interventions (increasing task complexity and modifying network capacity) induce an inverse or "contravariant" relationship between weight-space and dynamical degeneracy, while others (structural constraints) tend to affect both dynamical and weight-space degeneracy in the same direction or "covariantly" (Summarized in Table 1). Overall, the degeneracy of the solution space is primarily controlled by two key factors: the task complexity and the representational capacity of the networks. Among the manipulations to control degeneracy in RNNs, increasing task complexity and adding auxiliary losses directly alter the task complexity while representational load and adding structural constraints directly impact the representational capacity of the networks.

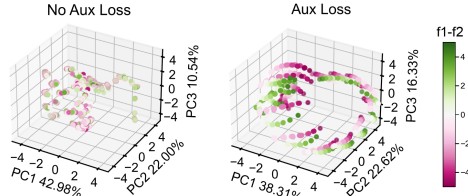

Figure 7: **RNNs learn more structured dynamics with auxiliary losses:** In Delayed Discrimination tasks trained with auxiliary losses (e.g., an auxiliary output $f_1 - f_2$ in addition to main output $sign(f_1 - f_2)$), RNNs neural dynamics or hidden states develop more sophisticated representations to track task-variables with 1-D manifolds (right) rather than just point-attractors (left).

## 4 RELATED WORK

Our approach extends a previously proposed measures of task complexity (Meister, 2022), which defined a task as a mapping from states to actions, and quantified task complexity as the entropy of all possible state-action mappings. Here, we regard tasks as predefined time-series of inputs and outputs rather than in terms of state-action mappings. By taking into account the temporal dynamics explicitly, we quantify the statistical complexity of the input and output time-series using Shannon entropy. Meister (2022) focuses on evaluating task-complexity from a learning-theory perspective: specifically, the amount of information an animal needs to acquire a perfect mapping between states and actions. In contrast, we focus on the representational capacity and the constraints imposed by the processing demands of a task. Our measure directly captures the amount of information that must be "carried" through the neural dynamics to produce the desired output. Our measure of task complexity is also consistent with the notion of *Neural Task Complexity* proposed by Gao et al. (2017), which provides an upper bound on the dimensionality that neural trajectories can explore during a task. We have demonstrated that within each type of task, variants with higher output complexity require higher dimensional representations (Figure 2D).

We further show that the output complexity of a task directly shapes the degeneracy of solutions found by RNNs. While previous work speculated that the degree to which neural dynamics are

attractor-heavy may influence degeneracy (Turner et al., 2021), we have quantified the "attractor-ness" of our tasks and found that it alone does not account for the different levels of dynamical degeneracy across tasks (see Appendix B). Moreover, as we modified the task characteristics to study its effect on degeneracy, we observed that degeneracy remains invariant under certain transformations of the task, e.g., changing the delay duration in Delayed Discrimination or altering the environment size in Path Integration tasks. In our framework, these transformations do not affect the output complexity of the task and thus leave the degeneracy unchanged. In the language of complex systems theory, these factors (e.g., delays) are "sloppy dimensions" of the task with respect to degeneracy. In contrast, factors that directly modify output complexity (e.g., adding channels, changing network size) are "stiff dimensions" that significantly control and modulate degeneracy of solutions (Gutenkunst et al., 2007b;a; Daniels et al., 2008; Kepple et al., 2022).

Our findings are also consistent with the *Contravariance Principle* from Cao & Yamins (2024): the harder a task, the more constrained the system that solves it will need to be. Another recent study by Huh et al. (2024) hypothesizes that AI models seem to be converging on the same representation spaces independent of modality, because more constraints from the massive-scale data make degenerate solutions less likely. Notably, the above related findings are based on feedforward networks for visual object categorization and Transformer networks for language modeling, respectively. Here, our study demonstrates that these ideas also generalize to recurrent networks/RNNs.

## 5 DISCUSSION

Our paper presents a unified approach to quantifying and controlling the degeneracy of solutions found by task-trained RNNs across behavioral, neural dynamical (or hidden states), and weight-space levels of analysis. We simulated and analyzed a task suite of broad interest to ML and neuroscience, yielding several novel insights.

First, information content in inputs and outputs of different tasks, which we use to quantify task complexity, emerges as a crucial determinant of degeneracy. This perspective helps reconcile previously conflicting observations on degeneracy across a range of tasks and measures. Second, across different tasks, as task complexity increases, we observe a robust inverse relationship between neural dynamical and weight degeneracy. As a corollary, while harder tasks lead to more consistent neural dynamics across task-trained networks, their underlying recurrent weight matrices are more degenerate/variable.

Third, our study exemplifies several strategies for controlling solution degeneracy, including manipulating task complexity, adding auxiliary loss functions, adjusting network capacity, and structural constraints on recurrent weights 1. These prescriptions will allow researchers to tailor the level of degeneracy of networks to suit specific research questions (e.g., probing individual variability across animals in neuroscience experiments) or application needs (Bouthillier et al., 2021; Morik, 2005; Yang et al., 2022). Importantly, we observed that certain strategies–increasing task complexity or modifying network capacity—induce an inverse (contravariant) relationship between weight and dynamical degeneracy, while others—imposing structural constraints—tend to affect both types of degeneracy in the same or covariant direction.

While our study is based on artificial neural network models, some of the above strategies could apply to neuroscience experiments, (Howard, 2002) e.g., introducing an auxiliary sub-task during behavioral shaping of lab animals to learn different tasks may constrain the degeneracy of the solutions learned by the biological brain and manifest in a different representational load than in an animal trained using a "curriculum" without an auxiliary sub-task. We note that degeneracy is also ubiquitous in other biological systems (Golub et al., 2018; Prinz et al., 2004; Edelman & Gally, 2001); thus, we motivate future theoretical studies based on them.

In summary, our work provides insights into the factors that shape the solution landscape of task-trained RNNs. The framework we bridges differing perspectives in prior work, reconciling observations of both universality and individuality in trained networks in ML and neuroscience. Future research could delve deeper into the theoretical underpinnings of the observed relationships in biological and artificial agents, including neural networks, between task complexity, representational capacity, and degeneracy.

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

# A  TASK AND TRAINING DETAILS

## A.1  N-BIT FLIP FLIP

| Hyperparameter | Value |
|---|---|
| Optimizer | Adam |
| Learning rate | 0.001 |
| Learning rate scheduler | None |
| Max epochs | 300 |
| Steps per epoch | 128 |
| Batch size | 256 |
| Early stopping threshold | 0.0005 |
| Patience | 3 |
| Probability of flip | 0.3 |
| Number of time steps | 100 |

## A.2  DELAYED DISCRIMINATION

| Hyperparameter | Value |
|---|---|
| Optimizer | Adam |
| Learning rate | 0.001 |
| Learning rate scheduler | CosineAnnealingWarmRestarts |
| Max epochs | 500 |
| Steps per epoch | 128 |
| Batch size | 256 |
| Early stopping threshold | 0.05 |
| Patience | 3 |
| Number of time steps | 60 |
| Max delay | 20 |
| Lowest stimulus value | 2 |
| Highest stimulus value | 10 |

## A.3  SINE WAVE GENERATION

| Hyperparameter | Value |
|---|---|
| Optimizer | Adam |
| Learning rate | 0.0005 |
| Learning rate scheduler | None |
| Max epochs | 500 |
| Steps per epoch | 128 |
| Batch size | 32 |
| Early stopping threshold | 0.05 |
| Patience | 3 |
| Number of time steps | 100 |
| Lowest frequency | 1 |
| Highest frequency | 30 |

## A.4 PATH INTEGRATION

| Hyperparameter | Value |
| --- | --- |
| Optimizer | Adam |
| Learning rate | 0.001 |
| Learning rate scheduler | ReduceLROnPlateau |
| Learning rate decay factor | 0.5 |
| Learning rate decay patience | 40 |
| Max epochs | 1000 |
| Steps per epoch | 128 |
| Batch size | 64 |
| Early stopping threshold (2D) | 0.1 |
| Early stopping threshold (3D) | 0.3 |
| Patience | 3 |
| Number of time steps | 100 |
| Lowest frequency | 1 |
| Highest frequency | 30 |

# B CHARACTERIZING THE ATTRACTORNESS OF THE TASKS

Attractors are associated with stable states in a network's dynamics. To quantify how attractor-heavy a task is, we measure the **speed of change** in the RNN's hidden activities. Specifically, we calculate the average normalized difference between the hidden states at two consecutive time steps. This measure reflects the stability of the network's internal states over time, where slower changes indicate the presence of attractor-like behavior.

For each trial, given a time series of hidden state activities, we calculate the speed of change in the hidden activities as:

$$\Delta h = \frac{\|h_{t+1} - h_t\|}{\frac{1}{T}\sum_{t=1}^{T}\|h_t\|}$$

where $T$ is the total number of time steps in the trial. We then average $\Delta h$ across all trials in the training set to obtain a representative measure of the average speed of change in hidden activations for the task. A low $\Delta h$ indicates that the hidden state is changing slowly over time, suggesting that the network is in or near an attractor state where the hidden activations are relatively stable. Conversely, a high $\Delta h$ indicates more rapid changes in hidden activity, implying less attractor-heavy dynamics.

As shown in Figure A1, tasks like N-Bits Flip-Flop and Path Integration, which have the highest task complexity and lowest dynamical degeneracy, span opposite ends of the "attractorness" spectrum. Interestingly, Delayed Discrimination, which exhibits the highest dynamical degeneracy, shows an intermediate attractorness score. This suggests that the attractorness score, as measured by the speed of change in hidden activity, does not fully account for the dynamical degeneracy observed in the networks. Therefore, while attractor-like behavior influences network dynamics, it is not the sole factor determining dynamical degeneracy across tasks.

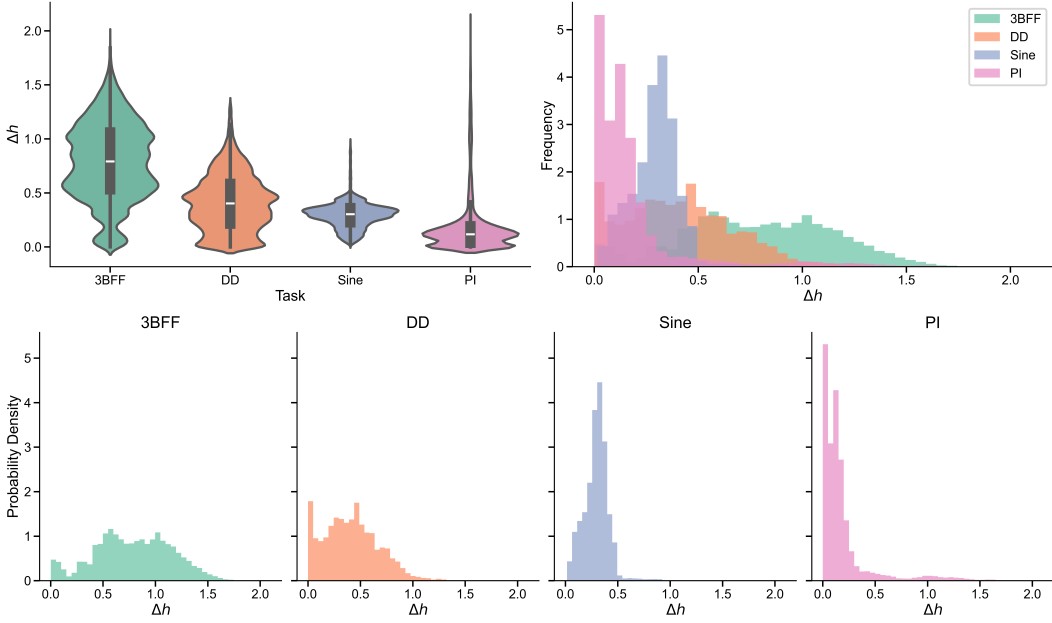

Figure A1: **Normalized speed of change in hidden state activities for each task:** (A) Speed of change in hidden state activity ($\Delta h$) for all trials in the training set plotted in violin plot for each task (B) Overlaid histogram of ($\Delta h$) for all tasks (C) Individual ($\Delta h$) for each task.

## C  DYNAMICAL SIMILARITY ANALYSIS (DSA)

Briefly, DSA proceeds as follows: Given two RNNs with hidden states $\mathbf{h}_1(t) \in \mathbb{R}^n$ and $\mathbf{h}_2(t) \in \mathbb{R}^n$, we first generate a delay-embedded matrix, $\mathbf{H}_1$ and $\mathbf{H}_2$, by sampling several hidden-state trajectories from each RNN. Next, for each delay-embedded matrix, we use Dynamic Mode Decomposition (DMD) (Schmid, 2022) to extract linear forward operators $\mathbf{A}_1$ and $\mathbf{A}_2$ of the two systems' dynamics. Finally, a Procrustes distance between the two matrices $\mathbf{A}_1$ and $\mathbf{A}_2$ is used to quantify the dissimilarity between the two dynamical systems and provide an overall DSA score, defined as: $d_{\text{Procrustes}}(\mathbf{A}_1, \mathbf{A}_2) = \min_{\mathbf{Q} \in O(n)} \|\mathbf{A}_1 - \mathbf{Q}\mathbf{A}_2\mathbf{Q}^{-1}\|_F$ where $\mathbf{Q}$ is a rotation matrix from the orthogonal group $O(n)$ and $\| \cdot \|_F$ is the Frobenius norm. This metric quantifies how dissimilar the dynamics of the two RNNs are after accounting for orthogonal transformations. We quantify Dynamical Degeneracy across many RNNs as the average pairwise distance between pairs of RNN neural-dynamics/hidden-state trajectories

| Task | Number of delay | Rank |
|------|-----------------|------|
| N-Bits Flip Flop | 30 | 1000 |
| Delayed Discrimination | 20 | 100 |
| Sine Wave Generation | 30 | 100 |
| Path Integration | 30 | 100 |

Table 2: Hyperparameters for DSA

## D  PERMUTATION-INDEPENDENT DISTANCE BETWEEN WEIGHTS

To quantify the dissimilarity between recurrent weight matrices in a permutation-independent manner, we use a variant of the Frobenius norm that allows for optimal row and column permutations. Given two RNNs with weight matrices $\mathbf{W}_1 \in \mathbb{R}^{n \times n}$ and $\mathbf{W}_2 \in \mathbb{R}^{n \times n}$, we define the Permutation-Independent Frobenius (PIF) distance as:

$$d_{\text{PIF}}(\mathbf{W}_1, \mathbf{W}_2) = \min_{\mathbf{P}_1, \mathbf{P}_2 \in \mathcal{P}(n)} \|\mathbf{W}_1 - \mathbf{P}_1\mathbf{W}_2\mathbf{P}_2\|_F$$

where $\mathcal{P}(n)$ is the set of permutation matrices of size $n \times n$, and $\| \cdot \|_F$ denotes the Frobenius norm.

This distance is calculated using an iterative coordinate-descent optimization strategy with multiple random restarts to avoid local minima. At each iteration, both the row permutation $\mathbf{P}_1$ or the column permutation $\mathbf{P}_2$ is optimized using a linear-sum-assignment approach as described in (Ainsworth et al., 2022).

