# OpenReview forum: "Measuring and Controlling Solution Degeneracy across Task-Trained Recurrent Neural Networks"
_ICLR.cc/2025/Conference — Submitted to ICLR 2025_

### Official Review · Reviewer_ecW8 · 2024-10-24

**Soundness:** 3
**Presentation:** 3
**Contribution:** 2
**Rating:** 5
**Confidence:** 2

**Summary:**

This paper proposes a unified approach to quantifying and controlling the degeneracy of solutions obtained by RNNs on several tasks by analyzing their degeneracy from three levels: behavior, neural dynamics, and weight space. They introduce some measures from information theory to quantify task complexity and conclude that increasing task complexity will reduce degeneracy in neural dynamics and generalization behavior, but increase the degeneracy in weight space. Based on these discoveries, they propose several strategies to control solution degeneracy in practice.

**Strengths:**

1. This paper proposes a unified approach to analyze the degeneracy of solutions obtained by RNNs on several tasks from several perspectives.

2. This paper incorporates certain measures from information theory to quantify task complexity. The authors conclude that an increase in task complexity will lead to a reduction in degeneracy in neural dynamics and generalization behavior. However, it will simultaneously increase the degeneracy in weight space.

3. Based on their analysis, the authors propose several strategies for controlling solution degeneracy in practical applications.

**Weaknesses:**

1. The results in this paper are empirical and lack of theoretical analysis. For instance, it would be better if the authors could add more discussion on the reason why increasing task complexity will increase the degeneracy in weight space. Maybe some theoretical analysis of some toy examples will help.

2. The strategies proposed for controlling solution degeneracy in practical applications are too general and may not be very useful in practice. For example, increasing task complexity reduces dynamical degeneracy and increases weight degeneracy, it is not clear which strategies we should use for specific tasks. More discussions on this aspect are welcome.

**Questions:**

Please see the Weaknesses part.

**Details Of Ethics Concerns:**

No.

---

### Official Review · Reviewer_F8Qm · 2024-11-03

**Soundness:** 3
**Presentation:** 4
**Contribution:** 2
**Rating:** 3
**Confidence:** 4

**Summary:**

The paper considers the effect of task complexity on “solution degeneracy”. The latter is characterized through examining how three measures behave as a function of task complexity: 1\) Similarity of the dynamical system between trained networks, 2a) difference in weights from initialization and 2b) pairwise similarity of weights between trained networks, and 3\) difference in generalization performance between trained networks.

### Recommendation
Overall, I think the paper is a valuable survey of the influence of task complexity on degeneracy and am looking forward to see it published. The main issue I see is with how expected the results are, depending on the way that it is regularized: An underconstrained system (as per less input-output channels) will be more variable due to unconstrained directions in the dynamics. If regularization in training, as is common practice, would turn out to remove for example DSA variability, I would not find the results sufficiently novel for ICLR. Until this point is clarified, I cannot recommend the paper to the conference.

**Strengths:**

- The paper is structured and clear.
- Jointly considering several degeneracy metrics (in particular DSA) and tasks is useful for the community.
- The tasks considered are diverse and rooted in the literature, giving a considerable degree of generality to the results.
- The related works section is thorough and bridges to the feedforward literature

**Weaknesses:**

(more important points first)

- **Regularization.** I could not understand from the paper whether a weight penalty was present when training on the task loss with BPTT. Weight decay for Adam is not specified, I assume it is the default, in case of which there would be some penalty. Was this the case, and if not, how would the measures change if the penalty got increased? For a stronger penalty, I would expect that the DSA metric shows less variation, as presumably noise directions that give rise to dynamical variability would then be penalized to decay. This would make the paper's finding much less interesting to me, because we already know that controlling task-unrelated dimensions is useful. I wonder for example whether Fig. 7 would be induced by strong regularization, biasing the network towards the presumably simpler line attractor solution.
- **Expectedness.** As the authors state in the discussion, most of the results can be understood with the following argument: As task entropy increases, the network parameters are increasingly constrained. Hence, there are less parameters that are unconstrained and give rise to degeneracy: Effectively, the network has now to solve $N$ tasks in parallel, but has the same representational capacity.
  This makes me wonder how the metrics change when task complexity is not scaled up by replicating the same task over channels, but changing the complexity of the function that needs to be learned? As a very simple and probably too naive example, how would the network behave if task complexity was controlled by the number of Fourier modes in the target signal? It seems that the paper identifies a specific kind of task complexity whose generality is questionable: The proposed measure is not rooted in the literature apart from a loose connection to information theory.
- **Weight degeneracy.** I find the metric not well motivated. Why should I expect a pair of networks that is non-degenerate (i.e., similar in some way) to have small $d\_{PIF}$? Would I not rather consider an (orthogonal) similarity measure on matrices, for example $d\_{sim} \= min\_O ||W\_1 \- O^{-1} W\_2 O||\_F$, where $O$ is an orthogonal or general invertible matrix. This is exactly the $d\_{procrustes}$ the authors discuss for DSA. I expect that using this metric on the weights would essentially make it the DSA metric. It would be especially interesting if that changes Table 1\.
- Do the authors have a hypothesis why change weight degeneracy depends on control paradigm?
- **Relevance.** Why should we care about whether a networks solution is degenerate or not? I understand that it is interesting descriptively to know a connection between task parameters and degeneracy metrics, but I am not sure where I would need this knowledge.  The paper would benefit from discussing this.
- **Controllability.** It is hard to generalize from the findings for controlling degeneracy from the DD task only. For example, it is hard to say what an auxiliary loss term would look like for the other tasks, possibly involving some amount of engineering.
- **Generality.** Do the authors suspect that there a tasks where the observed trends do not hold?
- No code is supplied, limiting transparency and reproducibility.

Again, I like the paper and think it is a valuable contribution, but these points make me question its novelty and generality.

**Questions:**

Please see the questions in the Weaknesses before.

---

### Official Review · Reviewer_hr4g · 2024-11-04

**Soundness:** 2
**Presentation:** 3
**Contribution:** 2
**Rating:** 5
**Confidence:** 4

**Summary:**

This paper presents an approach to explore & control the solution space of task-trained vanilla RNNs. They provide an information theoretic approach to measure task complexity and provide empirical results on its correlation with behavioral, weight space and neural dynamic degeneracy. Lastly, the also provide ways in which such degeneracy of solutions can be controlled. Their results are performed on four neuroscience inspired tasks.

**Strengths:**

The proposed method provides a unified approach to understanding the wide range of solutions in task-trained RNNs. The authors present multiple reasons/possibilities for these solutions (i.e due to behavioral, weight space or neural dynamic variability), and quantify its relationship with the amount of information present in each task. The information theoretic approach provides a way to categorize more complex v/s less complex tasks, which is then used to observe trends with other covariates. The authors further increase same task complexity by increasing the amount of information associated with the task (by adding independent channels). Their results suggest a similar trend in complexity with weight space degeneracy, and an inverse trend for neural dynamics and generalization. Lastly, they use these insights to increase/decrease the solution-space associated with tasks.

**Weaknesses:**

I have some main concerns/weakness which I will outline here with specifics under questions.

1. The authors present an interesting approach to quantify task-complexity using entropy. However, as detailed in the paper, this is very closely tied to the probability encoding the underlying generative process for each task. Specifically, while adding channels increases the information, a more robust measure would be changing the probability of the generative process (for instance the probability of a bit flip in a single channel). While in principle adding channels increases information associated with tasks, it's unclear if this is representative of the "randomness of the task".
2. The authors discuss ways to control solutions associated with task-trained RNNs. Other than increasing task-complexity and noting it's trend associated with weight and dynamical degeneracy (presented in this paper), the other four methods have been previously observed. Further, the authors do not provide any theoretical or discussion regarding how this relates to the method presented in this paper. Can the authors expand on the novelty of this section?

**Questions:**

1. As hinted at in the weaknesses, it's unclear if this measure relates to the true underlying generative process. Can they authors provide additional details on the generative processes' of these tasks (i.e where the uncertainty is, for instance in the delay period, or the probability of the bit-flip) and instead provide increased/decreased complexity of tasks based on this, and if the central claims of the paper still hold?
2. As detailed in 3.1, entropy calculated is discrete, however some tasks are continuous, yet no detail is provided on how binning can affect the final measure of H_{task}. Can the authors provide some results on this, and how this might change the 3 clusters of tasks they observe?
3. Can the authors provide any justification for why a joint/conditional/marginal entropy on input & output wasn't used?
4. Fig 2: while the general trend of more PCs are needed for more complex tasks, how can this trend be reconciled across tasks? For instance DD (high input, low output) needs more PCs than the most "complex" n-bit task. Similarly, the opposite is observed for path integration (fewer PCs) and n-bit which have similar complexity measures.
5. Fig 2: For panel E can the authors use either % or decimals consistently for variance?
6. Fig 3: Can the authors provide some justification or discussion for the trends observed, as the current writing goes it is unclear why such a trend makes intuitive sense. For instance, why does the output task complexity provide this trend v/s the input task complexity?
7. Fig 3: While the trend stated holds for same tasks with higher complexity, can the authors provide clarification on why between tasks this isn't observed? For instance sin-N and n-bff have similar output complexity measures but vary on DSA values. Can the authors provide some insight/clarification on this?
8. It appears the authors use different hidden dimensionalities in RNNs between tasks (i.e 64 v/s 128) while showing how their measure of task-complexity relates with degeneracy. As discussed in 3.3 this influences the capacity of the network. Can the authors provide a justification for why this was done and how the trends remain the same/change if all networks have the same dimensionality?

---

### Official Review · Reviewer_UQ1T · 2024-11-04

**Soundness:** 2
**Presentation:** 3
**Contribution:** 1
**Rating:** 3
**Confidence:** 3

**Summary:**

The paper provides ways of how to measure the behavior, dynamics, and weight-space degeneracies in RNNs, and then they provide methods to control these degeneracies. They show their findings using various tasks.

**Strengths:**

The authors tried their method in various tasks and settings.

The authors goal of measuring and controlling the degeneracy across different scales is very important and I think an important question.

The figures are very explanatory and the paper is written well, easy to follow.

**Weaknesses:**

"We chose this measure because it directly reflects the amount of information present in the task’s inputs and outputs, which the network must process and represent through its hidden state (Tishby et al., 2000)." In the experiments done in paper, the dimensions of the RNNs are most of the time higher than the dimensions of your input and output. I do not understand how does this method (of measuring the entropy of input and output _separately_) works, in the case when the goal of an RNN is to reproduce its input? Say one presented a highly complex input signal. And since the goal is to reproduce, then the output signal is also highly complex. But, the RNN only needs to implement an identity function, but the method suggested would classify this as a difficult task. I think this clearly demonstrates the problem of taking into account the input and output signal separately. At the end of the day, the goal of an RNN is to _map_ its inputs to outputs, so to the best of my understanding, there lies a significant problem in this approach. Since authors base all of their claims to this methods, I am highly confused and would like to have a discussion about this.

Controlling methods are only tried in one task but the results are written in general form. The claims in the paper, I believe could only be made if the authors have tried these methods on different tasks.

In the task settings in the paper I think it is true that increasing the number of channels makes the task more difficult, but I don't think it is always the case. Consider the delayed match-to-sample task. One can either have two different input channels to represent your signals or one. I think in this case representing with two channels makes the task easier. Does this show that this method is _not_ task agnostic?

Fig 4A, third column does not support your claim but you did not discuss it.

Auxiliary loss: I do not believe the defined auxiliary loss is in fact _auxiliary_ in the delayed discrimination task. When the task of a RNN is to output f1-f2 together with sign(f1-f2), I believe the main task becomes the former one.

**Questions:**

How does your permutation-independent distance contrast with the distance provided in Generalized Shape Metrics on Neural Representations paper Eq. 6?

It is written that "within each task, larger marker size and less opacity indicate task variant with higher complexity." Did the authors mean larger markers size and _more_ opacity? If not, I don't think your figure is interpretable.

For out of distribution tasks: What was the reasoning behind showing the CV but not mean error? Can you also show mean error and standard deviation separately?

What is the reasoning behind doubling the delay period or entire trial for OOD part?

---

### Official Review · Reviewer_Fhtp · 2024-11-04

**Soundness:** 2
**Presentation:** 4
**Contribution:** 3
**Rating:** 5
**Confidence:** 5

**Summary:**

Summary
The authors study the variability in RNNs across four tasks: a flip-flop task, a delay discrimination task, a sine wave generation task, and a path integration task. For each of these tasks, the authors propose a way to vary the task complexity, as measured by the entropy of the inputs and outputs. They found that the average DSA pairwise distance between 50 RNNs trained to reach a certain performance threshold, decreases as the task complexity increases. On the other hand, the degeneracy of the RNN weights, measured with a permutation invariant Frobenius norm, was found to increase with task complexity. RNNs trained on more complex tasks also showed decreased variability in their outputs when evaluated on the original tasks with longer time durations. Additionally, the authors showed that the weights and dynamics degeneracy varies depending on the task loss, the network size, and the weight regularization method.

**Strengths:**

* The study addresses an important question on diversity and degeneracy in RNNs.
* The authors investigate this question across multiple levels: behavior, dynamics, and connectivity, and across multiple settings: different loss functions, network sizes, and weight regularization methods.
* The paper is clearly structured and easy to follow.

**Weaknesses:**

Weaknesses
It seems that the central claim of the paper is: “while harder tasks lead to more consistent neural dynamics across task-trained networks, their underlying recurrent weight matrices are more degenerate/variable.” (Section 4). However, this statement may be overly simplistic and may significantly depend on factors that were not considered in the proposed work:
(a) Dependence on the initial weight distribution of the RNNs
(b) Dependence on the specific distance metrics used to quantify dynamics and weights degeneracy
(c) Dependence on the specific notion of complexity

(a) Prior work has shown that the initial weight distribution can have a large influence on the training dynamics and the solutions found by the RNNs. However, there is little information on how the RNN weights were initialized and there is no analysis on how the main claims may depend on their initial distribution. In particular, the decrease in dynamics degeneracy with task complexity may not hold for RNNs initialized with small weights with values close to zero.

(b) The authors measured weights degeneracy with a stricter Frobenius norm than the dynamical degeneracy. The norm used for DSA is invariant under orthogonal transformation, which is less strict than the permutation invariant norm used to quantify the weights degeneracy. The authors found that dynamics degeneracy decreases with task complexity but weights degeneracy increases. How much does this result depend on the specific norm and its invariance class used to measure degeneracy? Would the weights degeneracy decrease with task complexity when considering a similar matrix norm than for the dynamics degeneracy, i.e. orthogonal transformation invariant Frobenius norm?

(c) The authors quantify task complexity as the entropy of the task inputs and outputs. However, this captures only a specific aspect of the task complexity. For example, as stated in section 4:  “Moreover, as we modified the task characteristics to study its effect on degeneracy, we observed that degeneracy remains invariant under certain transformations of the task, e.g., changing the delay duration in Delayed Discrimination or altering the environment size in Path Integration tasks.”. It would be really interesting to see these results because increasing the duration of the delay period or increasing the size of the environment can be seen as ways to make the tasks more complex, in the sense that it would probably take longer for the RNNs to learn the tasks. The amount of training required to learn the task is another metric that can be used to measure task complexity. It would be interesting to study how it relates to the entropy based task complexity considered by the authors.

Measuring task complexity as the entropy of the task inputs and outputs implies that it is invariant under shuffling of the timesteps, which completely changes the task structure. Does the relationship between degeneracy and task complexity still hold when considering time shuffled variants of the tasks?

**Questions:**

Questions
Major questions:
* What is the distribution used to initialize the weights of the RNNs? How does the initial weight distribution affect the results? In particular, does the DSA distance still decrease with task complexity when initializing the RNNs with small weights close to zero?
* Why use a Frobenius invariant norm to measure weight degeneracy instead of an orthogonal invariant Frobenius norm as measured in DSA?

Additional questions:
* In general, how does the degeneracy evolve during training? Does it increase or decrease with training? This might depend on the weight initialization.
* It would be interesting to look at other more commonly used similarity measures than DSA such as CKA or Procrustes distance.

---

### Author Response · Authors · 2024-11-27

We want to thank all reviewers for their very insightful and constructive comments on our paper. Your critiques have inspired us to explore additional experiments and analyses that we believe will enhance the depth and comprehensiveness of our work. As fully implementing these new experiments and incorporating the new results will require more time than the rebuttal period allows, we have decided to take a step back and focus on a more extensive revision of our paper. We look forward to resubmitting it in the future with your critiques addressed. Thanks again for the time and effort you put into reviewing our submission.

---

### Meta-Review · Area_Chair_vcHr · 2024-12-20

**Metareview:**

This paper proposes a unified information theoretic approach to quantifying and controlling the degeneracy of solutions obtained by RNNs on several tasks by analyzing their degeneracy from three levels: behavior, neural dynamics, and weight space. The authors use their measures to study and control training RNNs on four very simple, neuroscience inspired tasks: a flip-flop task, a delay discrimination task, a sine wave generation task, and a path integration task.
The reviewers point out several strengths of the paper including: 1) The study addresses an important question on diversity and degeneracy in RNNs. 2) This question is studied across multiple levels and across multiple settings. 3) The presentation is clear and easy to follow.
The reviewers also raised several questions, partly asking for clarification, partly pointing out possible flaws in the approach, and partly questioning the relevance of the proposed method.
The authors decided that these comments warranted additional experiments which they were unable to complete in time. Rather than engaging in a discussion they decided to just thank the reviewers for their valuable feedback.
Based on the reviews and the authors reaction I recommend rejecting this paper, and look forward to seeing an improved resubmission at a later point.

**Additional Comments On Reviewer Discussion:**

No discussion. See above.

---

### Decision · Program_Chairs · 2025-01-22

Reject